# Strategies for Applications of Oxide-Based Thin Film Transistors

Lirong Zhang [1,2], Huaming Yu [2], Wenping Xiao [2], Chun Liu [1], Junrong Chen [1], Manlan Guo [2], Huayu Gao [3], Baiquan Liu [3,*] and Weijing Wu [1,*]

1   State Key Laboratory of Luminescent Materials and Devices, South China University of Technology, Guangzhou 510640, China; loezhang@foxmail.com (L.Z.); liuchun6795@163.com (C.L.); c527675149@163.com (J.C.)
2   Shunde Polytechnic, Foshan 528300, China; 10112@sdpt.edu.cn (H.Y.); xwp556@126.com (W.X.); guomanlan@foxmail.com (M.G.)
3   School of Electronics and Information Technology, Sun Yat-sen University, Guangzhou 510275, China; gaohy7@mail2.sysu.edu.cn
*   Correspondence: liubq33@mail.sysu.edu.cn (B.L.); wuwj@scut.edu.cn (W.W.)

**Abstract:** Due to the untiring efforts of scientists and researchers on oxide semiconductor materials, processes, and devices, the applications for oxide-based thin film transistors (TFTs) have been researched and promoted on a large scale. With the advantages of relatively high carrier mobility, low off-current, good process compatibility, optical transparency, low cost, and especially flexibility, oxide-based TFTs have already been adapted for not only displays (e.g., liquid crystal display (LCD), organic light emitting diode (OLED), micro-light-emitting diode (Micro-LED), virtual reality/augmented reality (VR/AR) and electronic paper displays (EPD)) but also large-area electronics, analog circuits, and digital circuits. Furthermore, as the requirement of TFT technology increases, low temperature poly-silicon and oxide (LTPO) TFTs, which combine p-type LTPS and n-type oxide TFT on the same substrate, have drawn further interest for realizing the hybrid complementary metal oxide semiconductor (CMOS) circuit. This invited review provides the current progress on applications of oxide-based TFTs. Typical device configurations of TFTs are first described. Then, the strategies to apply oxide-based TFTs for improving the display quality with different compensation technologies and obtaining higher performance integrated circuits are highlighted. Finally, an outlook for the future development of oxide-based TFTs is given.

**Keywords:** oxide semiconductor; oxide-based thin film transistor; low temperature poly-silicon and oxide; display; integrated circuit

## 1. Introduction

Over the past decade, TFT technology has made huge progress [1–5]. For displays, TFTs have almost entirely been employed as the key technology in active-matrix flat-panel displays (AM-FPD) such as LCD, OLED, and LED [6–8]. Thanks to the technical maturity of TFTs, more and more novel displays, such as transparent, flexible, high resolution (8 k), and Micro-LED, have been coming into consumers' sight [9–13]. Wearable devices, full-transparent displays, and rollable televisions (TV) have also already been marketed and commercialized successively, making it possible to see displays everywhere. In addition, TFT integrated circuits replacing the traditional silicon complementary metal-oxide-semiconductor (CMOS) chip have drawn much attention [14–17]. Although the performance of the TFT device and the structures of its circuits are simpler than conventional CMOS, it has still been prospected for the TFT in the amplifier, oscillator, inverter, simple logic circuit, and some data transmitting and receiving modules in a large area or on flexible substrates. Moreover, some large area sensor systems with combination structures of TFT arrays and CMOS chips have been proposed, including sensor systems of light, finger, temperature, pressure, pH, gas, and bio [18–22].

Oxide-based TFTs have attracted the interest of many researchers in recent years. Compared with amorphous silicon (a-Si) TFTs and LTPS TFTs, oxide-based TFTs have unique advantages. The mobility of oxide-based TFTs is about 1~100 $cm^2 \cdot V^{-1} \cdot s^{-1}$, which is suitable for driving OLED. It has also been verified to drive Micro-LEDs recently. Because the s orbitals in the conduction band of the oxide semiconductor are usually overlapped with each other, the carrier mobility is less affected by the ordering degree of thin film materials. Therefore, oxide-based TFTs have good electrical uniformity. The process temperature of oxide-based TFTs is low and can be compatible with the a-Si TFT process, making it possible to fabricate on a flexible plastic substrate. It can also realize low-cost manufacturing since no ion implantation and crystallization equipment is needed. However, there are some inherent disadvantages in oxide semiconductor materials, such as being only n-type devices, shifting under negative bias, and having a sensitivity to light. Therefore, it is necessary to apply different structures of TFTs for suppressing the functional degradation and different innovative circuit topologies for compensating degraded parameters.

Different device structures have different properties and applications. The traditional bottom gate structures in oxide TFTs are the back channel etch (BCE) and etch stopper layer (ESL) which are often used in commercial displays. In order to reduce the parasitic parameters, top gate (TG) and co-planar (CP) structures are designed. A dual gate (DG) structure can increase the output current of fixed size TFTs. With the emergence of flexible display technology, the horizontal channel is vulnerable to damage with the increase of bending times and the reduction of the bending radius. In order to solve this problem, the vertical channel device is designed. However, the above structures are only n-type for oxide-based TFTs. As is known, in a complementary structure with NMOS and PMOS, the CMOS circuit can obtain high performance and low power consumption. Hence, fabricating the p-type TFT to realize the complementary circuit with oxide-based TFTs has drawn much interest. LTPO TFTs, with p-type LTPS and n-type oxide TFTs on the same substrate, can realize the complementary structure, making it a promising future application. With LTPO structure, TFT circuits can be further designed and improved. Some complicated circuits are presented in [23–26], such as timing controllers, common drivers, gamma circuits, digital-to-analog converters (DAC), and power-supply circuits onto glass or plastic.

In the field of FPD technology, LCDs are one of the earliest commercial FPDs in the world and are still the mainstream of the display market, including TV, computer, automobile monitors, laptops, pads, and mobile phones. Over the past decades, more and more LCDs with higher resolution and larger sizes were launched [27–29]. In 2012, the Sharp Corporation announced its first Indium Gallium Zinc Oxide (IGZO) TFT-LCD mobile phone which could receive a 1 Hz frame frequency when showing a static picture. This phone presented low power consumption due to the low off-current of IGZO TFTs. In 2013, SONY released the world's first curved LCD TV which made people aware that LCD technology that can also be curved. This TV was only 65-inches and had a resolution of 1080 p. Then in 2014, the second generation of LCD curved TV was promoted with 4 k resolution and 65/75 inch screen which met the needs of consumer groups. LCD is a passive light-emitting device and its response time is a millisecond, which limits its application in the high-resolution display. Therefore, how to improve the operating speed of LCDs is a hot research issue. Although LCD technology is the most mature flat panel display technology, there are still challenges in some special fields, such as a fully flexible display, fully transparent display, and wearable display.

A new display must replace LCD technology to overcome the obstacles and expand new applications. The flexible and transparent displays (OLED) employing oxide-based TFTs were then rolled out over the past 10 years [30–36]. Oxide-based TFT technologies including new device structures [37–40], compensation pixel circuits [41–44], and narrow bezel gate drivers [45–50] are employed in AMOLED to improve image quality and stability for commercialization. It is known that OLED is an active light-emitting device driven by its flowing current. It is important to maintain the stability of current density with

OLED lighting. Due to the degradation of TFTs and OLED, the traditional 2T1C pixel circuit is no longer suitable for driving OLEDs. Furthermore, when the threshold voltage and current mobility of TFTs and OLEDs are offset, the current flowing through them will change under the same data voltage, resulting in the degradation or nonuniformity of the display screen. Hence, it is very urgent to develop and design pixel circuits with compensation technology. There are two kinds of compensation methods, which are the most common compensation techniques, called internal and external compensation. In most cases, internal compensation is suitable for high-resolution and small-scale screens, while external compensation is often used for large-scale displays. With compensation technology, OLED-based products have been commercialized. However, as an organic light-emitting device, OLED degrades easily and has obvious deficiencies in lifetime and light efficiency.

In recent years, Micro-LEDs have been receiving increasing attention as promising candidates for next-generation displays, due to their outstanding characteristics such as higher color reproducibility, higher brightness, lower power consumption, faster response time, and higher reliability [51–54]. TFTs have been demonstrated to drive Micro-LEDs with analog and digital driving methods [55–57]. With the complexity of pixel drive technology increasing, internal and external compensation methods, pulse width modulation (PWM), and pulse amplitude modulation driving methods have been employed to solve the performance degradation of TFTs and improve the display quality.

In addition to the display field, TFT applications have been expanded to the field of analog and digital circuits. There has been a lot of concern aroused for employing oxide-based TFTs for integrated circuits (ICs) such as amplifiers [58–60], inverters [61], oscillators [62–67], and RFID/NFC tags [68–70], as well as digital circuits including logic circuits [71–74], level shifter [75,76], flip-flops [77], data processors [78,79], and other digital circuits [80–82]. The reported oxide-based TFT amplifiers have a potential prospect in large-area, wearable or flexible applications, and bio-potential monitoring systems. However, there are various design challenges for oxide-based TFT realizing operational amplifiers as only n-type TFTs are available for integration. First, the transconductance of metal oxide TFTs is not high enough because the electrical mobility of metal oxide is much lower than that of crystalline silicon. Second, it is difficult to obtain high output impedance in amplifier design due to the lack of a p-type TFT. The oscillator is the key element to support the clock in phase locking loop (PLL), analog-to-digital converter (ADC), general-purpose or domain-specific data processing, and sequential logical circuits [78,79,82,83]. A conventional diode-connected inverter is usually used to establish an oscillator since only unipolar oxide-based TFTs are available for integrated circuit application, bringing large power consumption. Some optimized structures of oscillators reported by [63,67], focused on solving these related issues. RFID/NFC tags are widely used in applied in logistics, transportation management, financial security, door access control systems, and biological signal communication systems. Whereas the low operating frequency of RFID tags of oxide-based TFTs is the common obstacle to popularization and application. Digital circuits, memory devices, and sensor systems of oxide-based TFTs have already been achieved, such as basic logic gate circuits, level shifters, D flip flop, and simple domain-specific data processors.

With the progress of electronic devices, the performance of oxide-based TFTs has been enhanced step-by-step and nowadays can satisfy most of the demand of real commercialization for flexible displays, wearable devices, thin-film integrated circuits, and sensor systems. LTPO, combination p-type SnO, or p-type carbon nanotubes (CNTs) with n-type IGZO TFTs have been studied, attracting both industrial and academic interest [61,84]. Compared to the unipolar device, LTPO could not only avoid a high leakage current, low driving current, and low gain but also be integrated as data processors or other more complex circuits on large-areas and flexible applications which could make for the potential for replacing silicon-based integrated circuits and systems on glass or plastic. Hence, research work engaging in the oxide-based TFTs' technology is focusing on all aspects of TFTs to

obtain better-performance devices including materials, structures, processes, stabilities, and applications.

In this review, the carrier transport mechanism of oxide semiconductor materials is simply introduced in Section 2.1. Several popular kinds of configurations of oxide-based TFTs are briefly presented in Section 2.2, including bottom gate (BG), top gate (TG), dual gate (DG), co-planar (CP), vertical, and hybrid structures (HS). Then, the strategies to apply the oxide-based TFTs, for improving the display quality with different compensation technology, are presented in Section 3.1; after that, the thin film integrated circuits and systems are highlighted in Section 3.2. Finally, an outlook for the future development of oxide-based TFTs is given.

## 2. Device Fundamental Concept and Conventional Configuration

TFTs are one of the special types of metal oxide semiconductor field effect transistor (MOSFET), which is fabricated into a thin film active semiconductor layer, dielectric layers, and metal on a glass substrate or plastic by a deposition and magnetron sputtering process, respectively. In this section, the oxide-based TFT fundamental concept is discussed first, and the six common configurations of TFT are introduced, including their structures and implementation.

### 2.1. Device Fundamental Concept for Applications

Compared to monatomic silicon, oxide semiconductors are more complex materials because they are built up from several chemical compounds and can contain more different kinds of defects and impurities. Since the electron traveling orbitals are different from covalent semiconductors (e.g., silicon), the saturation mobility of oxide-based is of 10–30 $cm^2/(V \cdot s)$. Hence, it is beneficial to comprehend the carrier transport mechanism of oxide semiconductor materials [85–93]. Indium zinc oxide (IZO), indium gallium zinc oxide (IGZO), and zinc (Zn) are replaced by indium (In), which generates excess electrons, resulting in most oxide semiconductors exhibiting n-type characteristics in ternary and multinary compound semiconductors. Because of the degenerate band conduction, there are spatial metal n$s$ orbitals with isotropic shapes composing the carrier transport paths in metal oxide semiconductors. Due to the spherical symmetry of the $s$ orbital, it is possible to offer conductor pathways through the overlap in distorted metal-oxygen-metal.

In the case of the doped oxide semiconductors (e.g., a-IGZO), In has the electron structure of (n−1) d10n$s$0 (n > 5), where the electronic channel would be generated by overlapping the adjacent 5$s$ electron orbit. The 5$s$ electrons are spherically symmetrical and insensitive to direction. Hence, the degree of ion alignment has little influence on electron transportation as the conduction band minimum (CBM) is contributed by the In 5$s$ orbital, rather than by Zn 4$s$ or Ga 4$s$. The mobility and effective masses of monocrystalline and amorphous IGZO are relatively close due to the similar bandwidth of the conduct band level [94] and the close effective mass of 0.18 me and 0.2 me [95]. However, the mobility of IGZO depends on the electron concentration, which can be explained by the osmotic conduction model [96,97]. In this model, the center energy ($\phi_0$) of a-IGZO gets closer to the Fermi level ($E_F$) with rising electron concentrations, leading to a narrower distribution width ($\sigma_\phi$).

### 2.2. Oxide-Based TFTs Configurations

Oxide-based TFT devices have a variety of structures that are being investigated. By fabricating a gate, an oxide semiconductor, and drain/source electrodes in different ways and different orders, six common types of device configurations can be formed, namely BG, TG, DG, CP, vertical, and hybrid structure (HS), as shown in Figure 1.

In Figure 1a, there are two types of processes called BCE and ESL. BCE processes are applied in most commercial applications (e.g., cellphone and TV) due to their simple photolithography process and low cost as shown in the top half of Figure 1a. In that process, oxide semiconductors are very sensitive during most of the wet etching and dry

etching processing. There is still a risk of damage to the semiconducting back-channel during the S/D electrodes etch [98]. To avoid these effects, the ES layer contained in the ES configuration could be formed and patterned upon the active layer to protect the channel during the etching of the S/D electrodes [99], as shown in the bottom half of Figure 1a. However, it makes one more process and increases the cost.

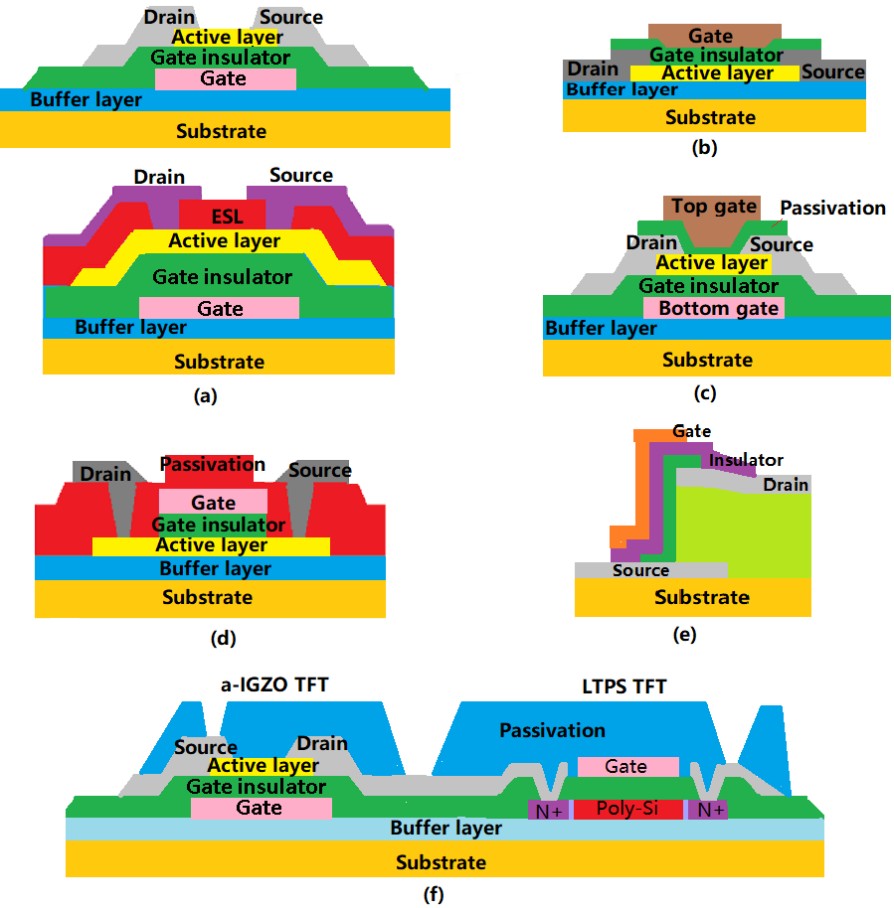

**Figure 1.** Typical device configurations of oxide semiconductors: (**a**) Back channel etch (BCE) and etch stop (ES) structure, (**b**) Top gate (TG) structure, (**c**) Dual gate (DG) structure, (**d**) Co-planar (CP) structure, (**e**) Vertical structure, and (**f**) Hybrid structure of LTPO.

Figure 1b shows the TG structure, where the on-top passivation is replaced by a gate insulator and electrode to protect the active layer [100]. Overlap capacitances between the gate and S/D electrodes can be effectively reduced by adapting the self-aligned gate process step. Therefore, the delay effect caused by parasitic capacitance could be significantly suppressed. Nonetheless, the stability of the negative bias illumination stress of the top gate structure TFT still needs to be improved.

By combining BG and TG (Figure 1a,b), DG structure with a top gate and a bottom gate in Figure 1c, presents a higher performance and stability through modulating channel carrier concentration, as shown in Figure 1c. Furthermore, the threshold voltage and the field-effect mobility of TFTs would be improved and the subthreshold swing (SS) reduced by providing TG bias. Researchers have investigated this structure for enhancing the compensation ability of pixel circuits and driving the current of gate driver circuits and frequency of oscillators in a limited substrate area [101].

However, it is observed that the overlap between the gate and source/drain electrodes causes a high parasitic capacitance and contact resistance, making a resistance-capacitance (RC) delay in TFT arrays and integrated circuits. To reduce this RC delay effect and improve the dynamic response speed, CP TFT is presented in reference [102], as shown in Figure 1d.

In CP TFTs, the overlap between the G and S/D electrodes is negligible and the contact resistance is reduced because of the gate on the same side as the semiconductor so that the RC delay effect can be remarkably reduced.

The last unipolar structure is the vertical structure shown in Figure 1e. The vertical structure has some merits that others do not have, such as achieving short channel, reducing the device area, and suppressing mechanical stress in a flexible substrate. When the cracks are generated by compressive and tensile strain in the channel layer, the carriers could still be transported in the vertical direction [103]. This structure has great potential for flexible application.

In order to achieve real complementary CMOS circuits, HS structure with an n-type TFT and p-type LTPS or carbon nanotube or SnO TFT as shown in Figure 1f, has received increasing interest from researchers despite the complex processing steps and high cost [61,104]. Employing HS structure into TFT circuits to resolve the common problems, such as waste of substrate area, non-full swing output, low noise margin level, and high power consumption has presented the possibility of replacing conventional silicon-based integrated circuits in some domain-specific fields.

### 3. Strategies for Applications of Oxide-Based TFTs

In this section, the recent applications in oxide-based TFTs are highlighted for displays, analog circuits, and digital circuits. In displays, the strategies for higher performance LCD, OLED, and Micro-LED, particularly in pixel driving, compensation circuits are emphasized. Then the recent analog applications of oxide-based TFTs including inverters, oscillators, amplifiers, and other circuits are introduced. Finally, the basic logic gates, flip-flops, and some digital processors and systems would be focused on.

*3.1. Display Technology*

3.1.1. LCD

LCD is the most mature and common display technology and has a long history of commercialization in the world, such as smartphone displays, portable mobile devices, and televisions. In 2018, Hara et al. from Sharp Corp. reported the first mass production of Indium-Gallium-Zink-Oxide TFT (IGZO-TFT) in the world. ESL structure TFTs were employed to achieve high resolution and a large-screen 8 k LCD with a gate driver in panel (GIP), and then AUO Corp. pushed out its first true 1G1D 75″8 k 4 k LCD with oxide TFTs without any additional compensation [105,106].

Since LCD is a voltage-driven device, a simple circuit that includes 1 TFT and 1 capacitance (1T1C) in a pixel can distort the LCD by voltage signal through the switching TFT; as a result, the LCD is relatively easier to be fabricated in large-area, as shown in Figure 2. When the scan signal arrives, the data signal will be stored in Cs and $C_{LC}$, noting that $C_{LC}$ is the capacitor of LCD. Meanwhile, as shown by the waveform change of SCAN and DATA signal in Figure 2, RC delay time and power consumption are critical focus areas rather than the degradation of TFTs. In high-resolution displays, high field mobility TFTs and Cu metal bus lines are employed to resolve the problems of high resistance and heavy capacitive load. Novel backlight technology is devoted to reducing the power consumption of LCDs, such as LED and QLED rather than the cold cathode fluorescent lamp technique. Additionally, the driving method of the pixels with changing their conventional features could also reduce the power consumption. Kim et al. proposed a new pixel circuit comprising LTPO TFTs with panel self-refresh technology to reduce the frame rate by adding a memory circuit to the pixel structure [107]. LTPO TFTs with low leakage current and high mobility can simplify the circuit. To improve the performance of LCDs, Jo et al. designed a 3-bit grayscale self-refreshing memory in pixel circuit with LTPO to reduce power consumption in global signal lines [108]. DG structure TFTs under synchronized bias on the top gate provides a larger current and steeper SS compared to the single-gate TFTs under the same channel width/length ratio. These superior properties of oxide-based TFTs allow the AMLCD to have high pixel density and high resolution [109]. However, LCD still has

some insurmountable difficulties like passive lighting, low response time, thicker devices, high power consumption, and flexibility compared with OLED and Micro-LED displays.

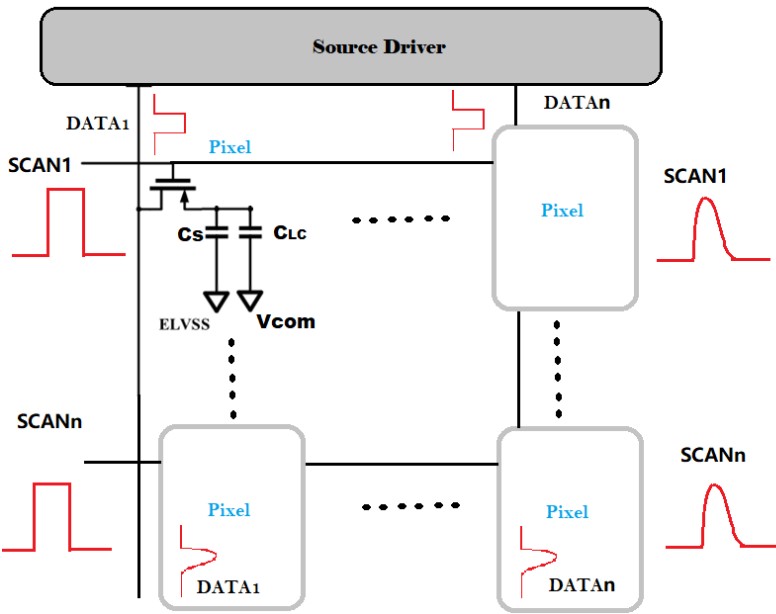

**Figure 2.** Pixel circuit and schematic diagram of an LCD panel.

### 3.1.2. OLED

Unlike LCD, OLED is a current-driven device that needs precise and steady current flowing through TFTs to make its own lighting [110–117]. Accompanied with many advantages like high luminous efficiency [118–122], fast response time [123–128], wide viewing angle [129–133], thinner devices [134–136], and low power consumption [137–139], OLED is praised as the next-generation display technology before Micro-LEDs appear. OLEDs driven by oxide-based TFTs have the obvious superiority that can realize full transparency and real flexibility, enabling a wide range of applications and commercialization. When using OLEDs on glasses or flexible substrates, some key technique points must be resolved, such as uniformity, quality picture, lifetime, and power [140–143]. In Figure 3a, a 2T1C circuit is employed to drive an OLED. When data signal $V_{DATA}$ reaches the gate of the driving TFT, the driving TFT works in the saturation region. The current flow through the device is expressed as.

$$I_{TFT} = \frac{1}{2}\mu c_{ox}\frac{W}{L}(V_{DATA} - V_{OLED} - V_{th})^2 \tag{1}$$

where $W$, $L$, $\mu$, $C_{ox}$, $V_{DATA}$, $V_{OLED}$, and $V_{th}$ are the channel width, the channel length, the field-effect mobility, the gate oxide capacitance per unit area, the data voltage signal, the threshold voltage of OLED, and threshold voltage of driving TFT, respectively. When TFT and OLED degrade, $V_{OLED}$ and $V_{th}$ will shift positive or negative to change the $I_{TFT}$, resulting in non-uniformity of the display. It is then necessary to implement the compensation technology into pixels. There are two kinds of compensation technologies to compensate for the degradation of TFTs and OLEDs, called internal and external compensation as shown in Figure 3b,c. Internal compensation is the early compensation technology in AMOLED, which compensates the threshold voltage shift of the driving TFT before transporting the current to OLED. The driving TFT is then kept working in the saturation region and the driving current becomes independent of the threshold voltage. In Figure 3b, one of the most common circuit structures for $V_{th}$ compensation of TFT is presented. When T1 is on, T3 is off, $C_{st}$ will be discharged through T1 and T2 until the voltage between node A and C

is close to $V_{th}$ of T2; T2 will then be cut off to stop discharging so that the $V_{GS}$ of T2 is $V_{th}$. When the data signal arrives, Equation (1) can be expressed as

$$I_{TFT} = \frac{1}{2}\mu c_{ox}\frac{W}{L}(V_{DATA})^2 \tag{2}$$

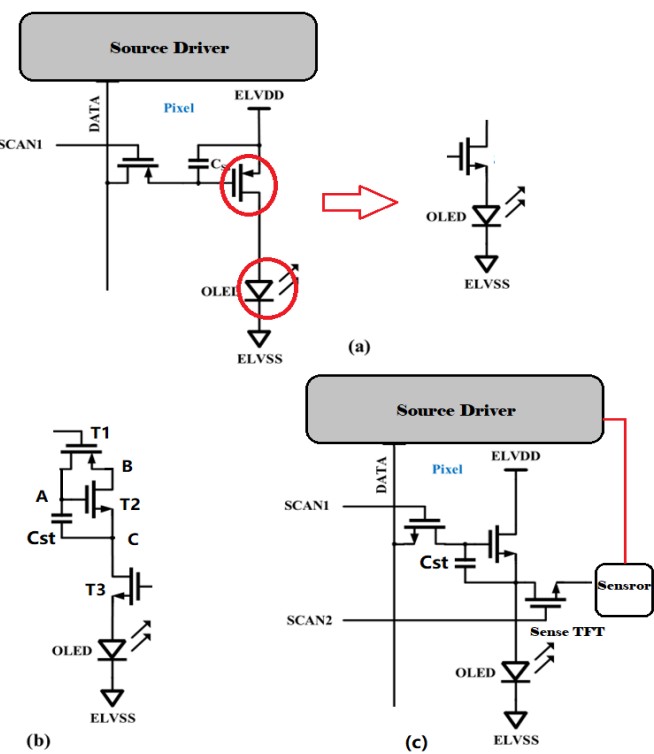

**Figure 3.** Degradation of TFTs and OLEDs and compensation technology: (**a**) Degradation in 2T1C circuit. (**b**) Internal compensation method. (**c**) External compensation method.

Ideally, the $V_{th}$ of T2 has been compensated, and the driving current is proportional to the data voltage which makes the OLED pixel luminance uniform. Note that the internal compensation just compensates the $V_{th}$ of T2, while the shift of $V_{OLED}$ still exists.

In Figure 3c, the schematic of external compensation technology is shown. An additional TFT called sense TFT is implemented to sense the voltage and current of driving TFT. Compared to sensing signals and data signals in the lookup table, the timing controller and source driver adjust the output data in real-time. Hence the $V_{th}$, $V_{OLED}$, and IR drop can be compensated.

In order to take full advantage of oxide-based TFTs and OLEDs, Wu et al. employed a threshold voltage one-time detection method to improve the pixel circuit speed. The $V_{th}$ detection period may occupy most of the whole programming time which generally reaches tens of microseconds, however, oxide-based TFTs have a lower off-current which can realize the threshold voltage one-time detection period after multiple frames rather than a detection for a frame as shown in Figure 4 [112]. Fan et al. proposed an AMOLED pixel that could compensate the $V_{th}$ of driving TFTs and OLEDs as shown in Figure 5 [113]. T4 is for OLED recovering when Scan1 is high. However, internal compensation is less effective for OLEDs and IR drop compensation.

For flexible AMOLED displays, TFT and OLED degradation induced by the strain of shape changing or thermal effect has been becoming a burning question. Kang et al. proposed a pixel circuit that could compensate for the variation of $V_{th}$ of TFTs and luminance reduction on a stretchable substrate for AMOLED in Figure 6 [114]. Dual gate structure with $V_{th}$-controlled by the top gate was adopted into T1. The reduction in luminance as display stretching can be compensated by the variation and transfer of charge in Cs and C2.

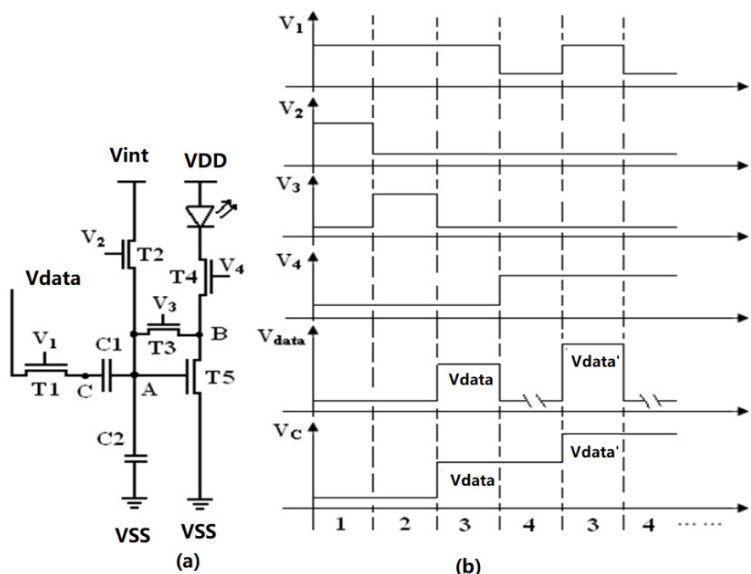

**Figure 4.** One-time detection method pixel circuit: (**a**) Schematic of the pixel circuit. (**b**) Timing diagram [112] (Reprinted with permission from ref. [112]. Copyright 2022 IEEE).

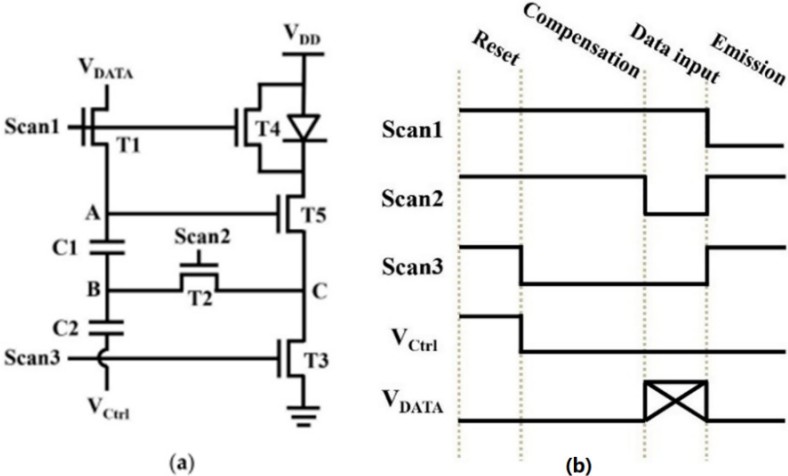

**Figure 5.** (**a**) Schematic of the pixel circuit. (**b**) Timing diagram [113].

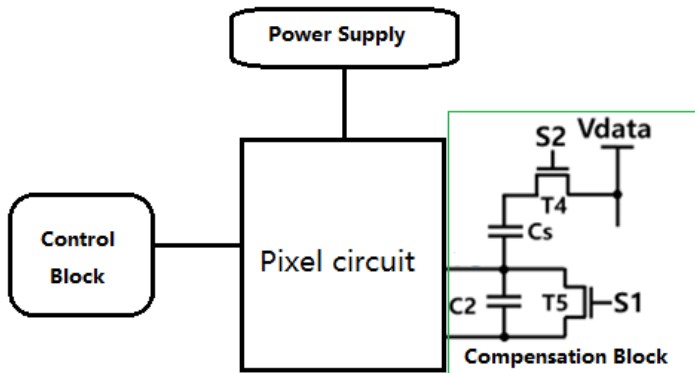

**Figure 6.** Schematic of the pixel circuit.

However, there will be non-uniformity in the characteristics of TFTs and OLEDs due to nuances in the manufacturing process, especially in large area displays. Internal compensation is just limited to compensating the $V_{th}$ shift of the driving TFTs and OLEDs,

while the mobility shift cannot be compensated. Therefore, external compensation must be introduced to compensate not only $V_{th}$ and mobility shift but also the degradation of the OLED and IR drop in large-area and high-resolution OLED displays.

Since the external compensation process is accomplished by a peripheral driver system, which includes ADC, DSP, CPU, and memory, the structure of the pixel circuit can be simplified as several TFTs and one capacitor, such as a 3T1C or 4T1C, plus a sensing line.

There are two kinds of external compensation methods that have been used in OLEDs, one is called real-time compensation and the other is non-real-time compensation. The real-time external compensation method is that the sampling signal feedback and pixel circuit programming process are performed at the same time. The feedback aging or non-uniformity data is immediately applied to the display data. The non-real-time external compensation is a design in which the sampling signal feedback and pixel circuit programming process are performed in different periods.

Kwon et al. proposed an external compensation approach for n-type TFTs panel with a 4T1C pixel circuit where the driver IC concludes gain amplifier, current integrator, ADC, and low pass filter (LPF) as shown in Figure 7a [115]. This method could not only compensate for the degradation of TFTs and OLEDs but also mitigate the effect of thermal, shot, and flicker noise, which has a potential application in large-size OLED applications. Shin et al. employed a real-time external compensation method on the enhancement of the luminance uniformity in large-size OLEDs based on a-IGZO TFTs by using external circuit and optical measurement methods [116]. The simplified schematic diagram is shown in Figure 7b. There are two steps in the compensation process; the first step is to compensate the $V_{th}$ and the mobility of the TFTs and the second step is to compensate for the luminance non-uniformity by optical measurement, making luminance uniformity 99% and creating a more than 10% improvement over the conventional method.

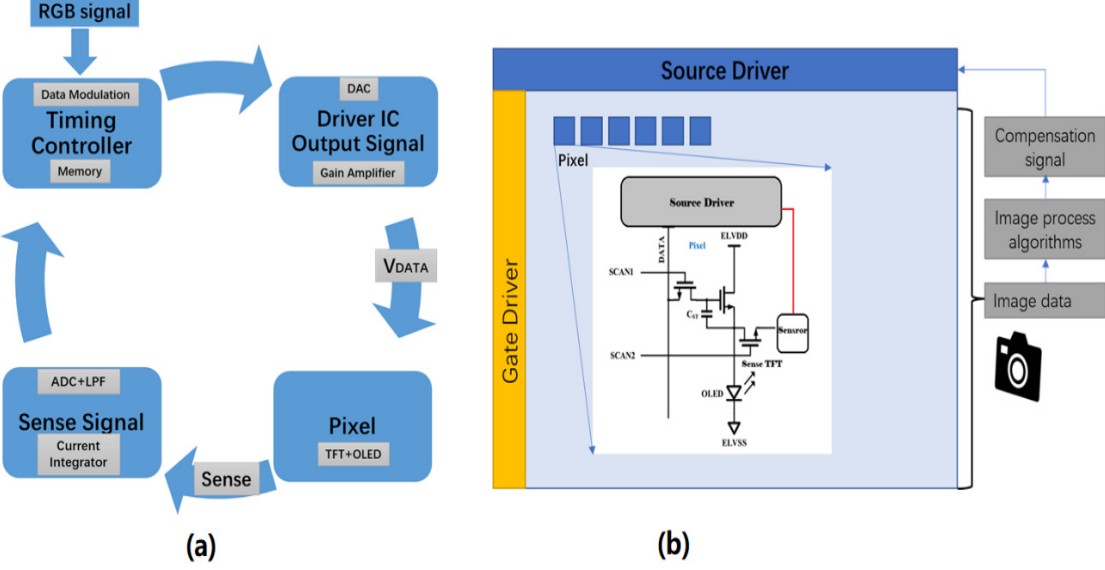

**Figure 7.** External compensation method: (**a**) Simplified block diagram. (**b**) External compensation approach.

In order to spend less time on the read-time compensation method, Lam et al. used a progressive compensation method [117] in which the sequence of display data is divided into K lows per segment. K pixel circuits in each segment were compensated by a peripheral compensation system before display data was input. A compensation method 14 times faster than the traditional method was achieved.

External compensation is a compensating generic technology, not only applied to oxide-based TFTs but also applied to LPTS, a-Si TFT [141]. However, it will bring high power consumption, as the process operates continuously, and an increase in cost, with

a need for additional compensation units. Recently LPTO TFTs for compensation pixel circuits have drawn much interest, due to high mobility and low off leakage current. Some researchers have reported LPTO pixel circuits to compensate $V_{th}$ of driving TFT and degradation of OLED as shown in Figure 8a–c [141–143], but they have been reserved for theoretical analysis and simulation verification. In most LTPO pixel circuits, LPTS TFT serves as the driving TFT for OLEDs due to its high mobility, while oxide-based TFTs serve as the switch transistors to keep the charge of each node of the storage capacitor. Aman et al. proposed a 6.39 in LTPO AMOLED with 403 PPI in Figure 8d which demonstrates the compensation of the pixel circuit [25]. Due to the process limitations, the LTPO technology display is only employed for mobile phones and portable devices.

### 3.1.3. Micro-LED

Micro-LEDs have been receiving much attention recently as a promising candidate for next-generation display technology, due to their outstanding characteristics such as higher color reproducibility, higher brightness, lower power consumption, faster response time, and higher reliability. These advantages enable Micro-LEDs to be used in all display scenarios including televisions (TVs), phones, wearable devices, augmented reality (AR), and so on.

However, there are two key problems for Micro-LEDs applied as a display. The luminous efficiency of Micro-LEDs will drop sharply in low current density and the wavelength of emitted light will shift with the current density. This causes a wavelength shift in the Micro-LEDs, resulting in a color shift and image distortion [145–148]. The pulse width modulation (PWM) driving method is expected to overcome the above problems because it modulates the grayscale by changing the emission time of Micro-LEDs with a constant operation current density. The PWM driving method can be divided into digital mode and analog mode.

The digital PWM driving method divides the frame time into multiple subframes and controls gray scales by switching subframes. It has the features of simple pixel circuits and complex driving timing compared with the analog driving method. The digital PWM driving method may be also applied in active-matrix organic LEDs (AMOLED) to reduce static power consumption by different subframe division methods.

TFTs backplane technology for Micro-LED displays will be compatible with the existing display processes, whose applications can cover mainstream consumer electronics such as watches, mobile phones, pads or tablets, and TVs. Until now, there have been few reported works on Micro-LED displays driven by a TFT backplane using the PWM method.

Our group, for the first time, has reported a Micro-LED compensated pixel circuit driven by the digital PWM method based on oxide-based TFTs as shown in Figure 9 [146]. Rather than the analog PWM driving method, the digital PWM method does not need the sweep signal, but the frame time must be divided into multiple subframes. As the result, the challenge comes from the number of subframes in a frame time. The more subframes, the less circuit time for compensation and displays. Hence, the digital PWM driving method may only be suitable for small size Micro-LED displays.

There are few pieces of research for the analog PWM method based on oxide-based TFTs. Oh et al. used LTPS-TFT for the analog PWM method and internal compensation, which is a valuable reference for oxide-based TFTs to realize the analog PWM method [147]. The hybrid driving method has also been reported in some research; Kim et al. combined the PWM and PAM method in which the PAM unit operates the external compensation of the driving TFT, while the PMW executes the internal compensation and driving process as shown in Figure 10a [52]. Ito et al. designed a current and PWM hybrid driving circuit where the driving current is lower than a certain level as shown in Figure 10b [54]. This kind of circuit is not suitable for reducing the wavelength shift of the emission light. A certain level may be defined according to the actual display situation. LTPO has been reported in Micro-LED displays which can simplify the structure of the circuit, reduce the

leakage current, hold the charge in storage capacitors, and maintain the current density flowing through the Micro-LED.

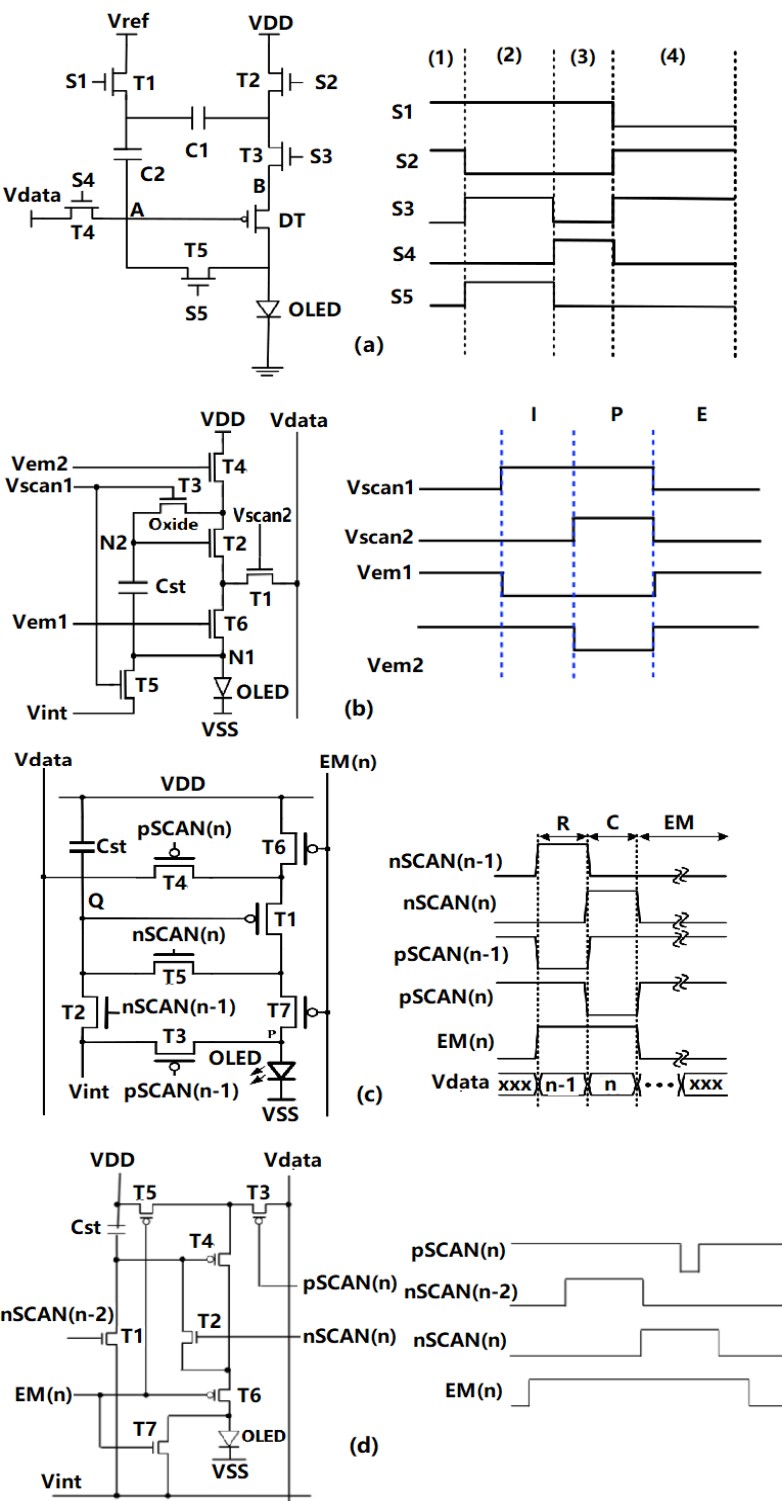

**Figure 8.** Four kinds of LTPO pixel circuits: (**a**) 6T2C with one driving LPTS TFT [142] (Reprinted with permission from ref. [142]. Copyright 2022 John Wiley and Sons), (**b**) 6T1C with one oxide TFT [141] (Reprinted with permission from ref. [141]. Copyright 2022 John Wiley and Sons), (**c**) 7T1C with one oxide TFT [143] (Reprinted with permission from ref. [143]. Copyright 2022 John Wiley and Sons), and (**d**) 7T1C with three oxide TFTs [144] (Reprinted with permission from ref. [144]. Copyright 2022 John Wiley and Sons).

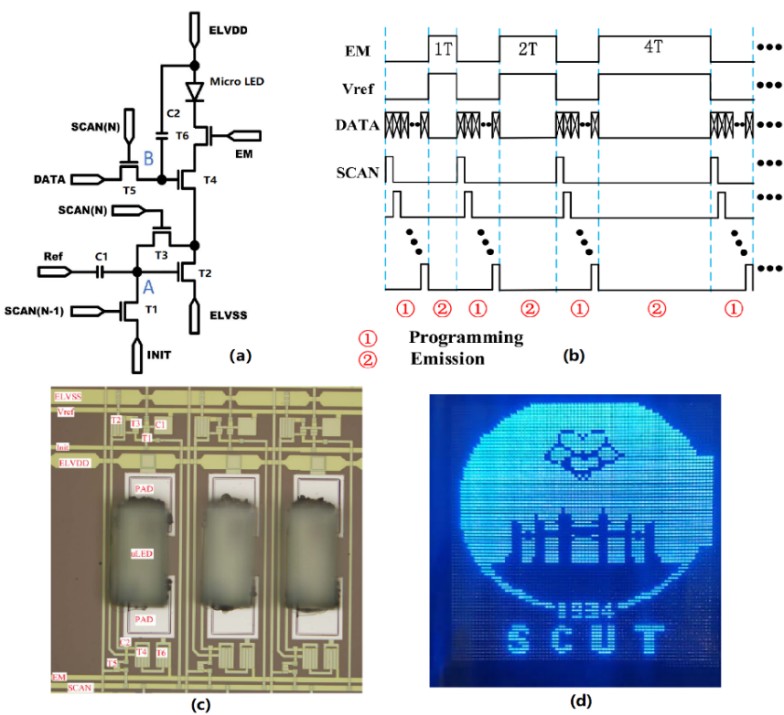

**Figure 9.** (**a**) Schematic of the proposed pixel circuit, (**b**) Timing diagram of the proposed pixel circuit, (**c**) Optical micrograph of the Micro-LED pixel circuit, and (**d**) Display of the Micro-LED [146] (Reprinted with permission from ref. [146]. Copyright 2022 IEEE).

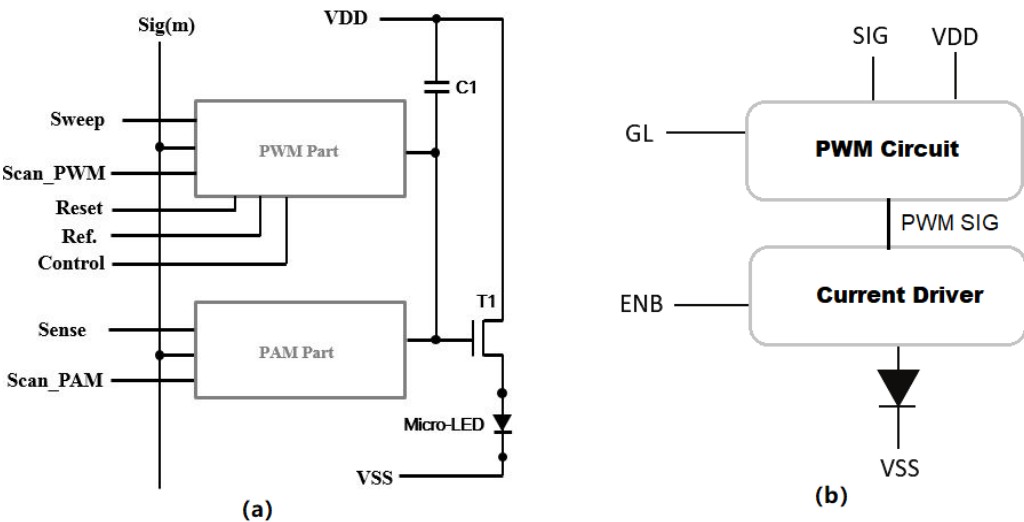

**Figure 10.** (**a**) PWM pixel circuit [52] (Reprinted with permission from ref. [52]. Copyright 2022 John Wiley and Sons). (**b**) Schematic diagram of current + PWM pixel circuit.

With the development of pixel circuits from LCD to Micro-LED, 1T1C to nTnC, and from simple voltage driving methods to hybrid driving methods, the requirement functions for circuits are increasing. Oxide-based TFTs play a key role along with commercial mass display production and flexible displays. The detailed structures of representative pixel circuits of oxide-based TFTs are described in Table 1. Further, oxide-based TFTs have broader applications besides display, like analog and digital circuits.

### 3.2. Oxide-Based TFT Integrated Circuits

Compared with CMOS devices, oxide-based TFTs have low performance, such as low mobility, larger area, and larger parasitic parameters, which makes it impossible to

integrate high-performance chips. However, it is suitable for large-scale productions and applications, such as large-area circuits and flexible circuits. In recent years, oxide-based TFTs have also attracted much attention in the field of biosensors. In Section 3.2.1, analog circuits are introduced, including simple inverting buffers, oscillators, amplifiers, and other circuits. The application of TFTs in analog circuits and its future development trend are introduced. In Section 3.2.2, the simple TFT logic gate circuits are shown. The flow is introduced from small scale integration (SSI), like triggers, to the very large scale integrated (VLSI) circuits, such as RFID chips applied in the Internet of Things and processors applied in the sensing field.

**Table 1.** Summarized performances for representative pixel circuits of oxide-based TFTs.

| Pixel Circuit | Structure of Pixel Circuit | NO. of Pixel Signal Lines | NO. of Supply Lines | NO. of Global Signal Line | Driving Method | Display Device |
|---|---|---|---|---|---|---|
| Reference [107] | 4T2C LTPO | 3 | 1 | 2 | Self-refreshing | LCD |
| Reference [108] | 4T2C | 2 | 1 | 4 | Self-refreshing | LCD |
| Reference [109] | 1T1C DG | 2 | 2 | 0 | Voltage program | LCD |
| Reference [110] | 5T2C | 3 | 2 | 0 | Internal compensation | OLED |
| Reference [111] | 4T2C | 3 | 2 | 0 | Internal compensation | OLED |
| Reference [112] | 5T2C | 4 | 2 | 0 | One time internal compensation | OLED |
| Reference [113] | 5T2C | 3 | 2 | 0 | Internal compensation | OLED |
| Reference [114] | 6T3C | 3 | 4 | 0 | Internal luminance compensation | OLED |
| Reference [115] | 4T1C | 3 | 2 | 1 | External compensation | OLED |
| Reference [116] | 3T1C | 3 | 2 | 1 | External compensation | OLED |
| Reference [117] | 3T1C | 3 | 2 | 1 | External compensation | OLED |
| Reference [146] | 5T2C | 4 | 3 | 0 | Internal compensation + digital PWM Method | Micro-LED |

### 3.2.1. Analog Circuits

Inverter

It is well known that the inverter reversing phase of the input signal to 180 degrees is usually employed in analog circuits, such as an operational amplifier (OA), clock oscillator, and buffer. The CMOS inverter is the core of almost all digital integrated circuits with its large noise margin, high input resistance, and low power consumption.

The oxide-based inverter plays an important role in flexible sensors, displays, and rectifiers for RFID and OA. There are six kinds of common structures of inverters, of only n-type oxide-based TFTs, as shown in Figure 11. A flexible IGZO inverter circuit with diode-connected transistors is proposed in [149], which showed decent performances (voltage gain > 1.5). However, the static power consumption is hardly neglected when the pull-down TFT turns on, especially for large-area use. The N-type zero-$V_{gs}$-load inverter has a poor fan out capability when the pull-down TFT turns off. The dual-gate inverter could reduce the static power consumption by shifting the $V_{th}$ of pull-up TFTs through the top-gate bias voltage when the pull-down TFT turns on, but it needs one more control line. N-type diode-load pseudo-CMOS logic could also optimize the power by reducing the size of diode-load TFTs. A bootstrapped NOT gate could improve the conduction of pull-up TFT through the coupling effect of capacitance, therefore, the size of pull-up TFT could be optimized to decrease current when the pull-down TFT turns on. The output level is prompted to GND or low level.

However, it is hard for one type or pseudo-CMOS inverters to achieve rail to rail output voltage. LTPO, oxide-based TFT vs. CNT [150], and p-type vs. n-type oxide TFTs are introduced to realize the higher gain and lower power inverters. All hybrid method inverters have simpler structure-CMOS. Figure 12a,b shows the LTPO inverter with a high

gain of 264 and 114.28 VV$^{-1}$, respectively [151,152]. Yuan et al. proposed oxide-based complementary inverters to achieve high gain and low power by using p-type SnO and n-type IGZO TFTs as shown in Figure 12c [61]. When the VDD is applied to 3 V, the gain can reach 226 VV$^{-1}$.

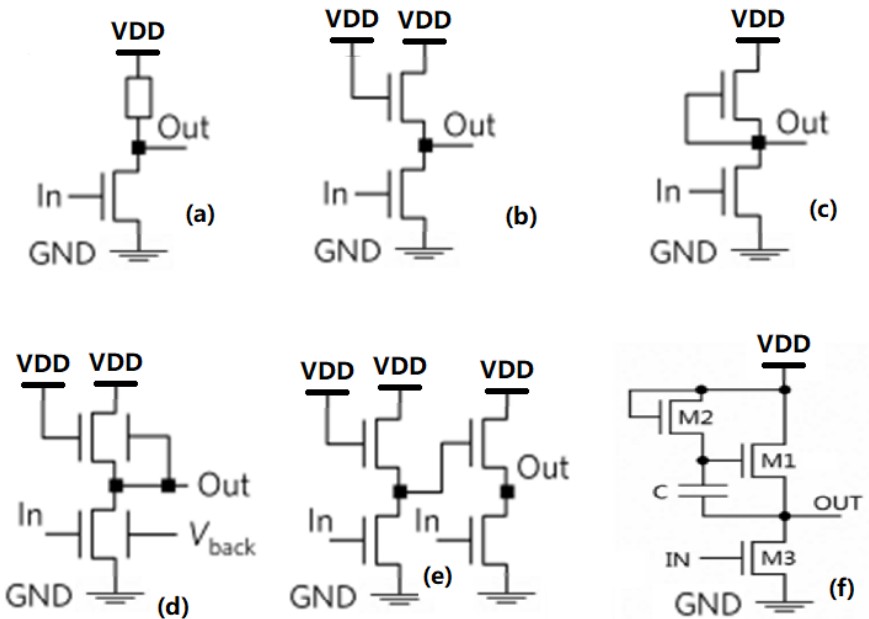

**Figure 11.** (**a**) Resistive-load logic inverter, (**b**) Diode-load logic inverter, (**c**) Zero-V$_{gs}$-load logic inverter, (**d**) Dual-gate n-type diode-load logic inverter, (**e**) n-type diode-load pseudo-CMOS logic inverter, and (**f**) Bootstrapped NOT logic inverter.

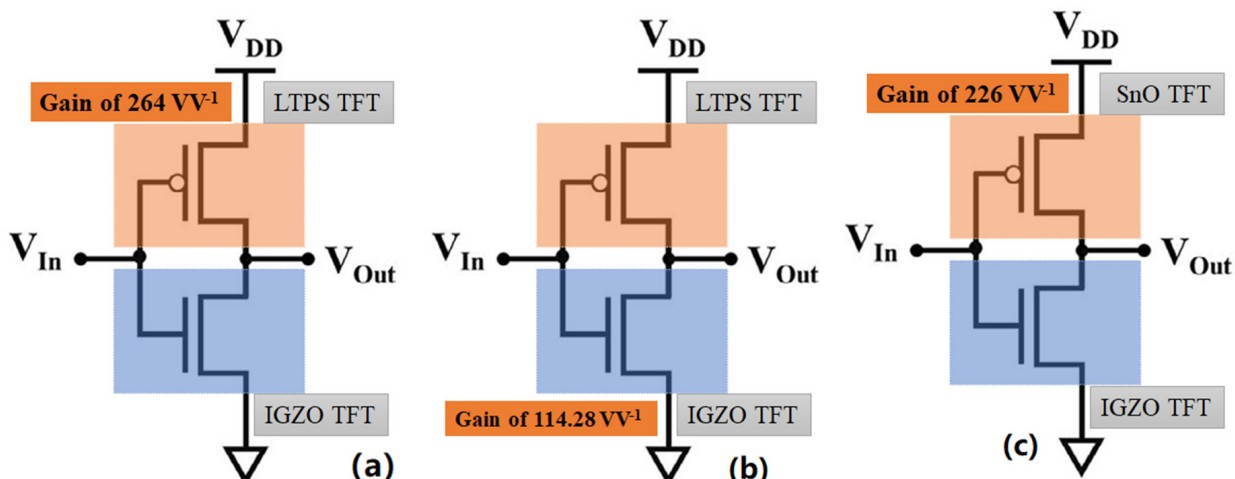

**Figure 12.** Inverter with LTPO: (**a**) Inverter of 264 VV$^{-1}$ gain, (**b**) Inverter of 114.28 VV$^{-1}$ gain, and (**c**) Inverter of 226 VV$^{-1}$ gain.

Oscillator

There are two kinds of clock generation circuits: a ring oscillator (RO) and a LC voltage-controlled oscillator (VCO). Due to the advantages of simple structure, compact size, wide tuning frequency range, and scalability of generating a multiphase clock, a RO is more attractive to the future flexible electronics product compared with a LC VCO [63]. Nowadays, researchers have attempted to fabricate high-performance ROs, realized by connecting an odd number of inverters into a ring, with TFTs along with the development of TFT technology.

Thereinto, the conventional diode-connected inverter and zero-VGS load inverter are usually used to construct the RO since there are only n-type MO TFTs available. A diode-connected transistor serving as a pull-up transistor of the inverter will lead to large power consumption when the input signal is high. Moreover, the threshold voltage of oxide-based TFTs is generally around zero voltage, which makes it difficult to completely shut off or turn on the pull-up or pull-down transistors in a conventional inverter circuit. Yang et al. presented a high-speed oscillator with AlInZnSnO and InZnO double-layer oxide TFTs, which can reach 296 kHz at VDD = 20 V, but the peak-to-peak voltage is only about 10 V [64]. A pseudo-CMOS inverter is used in ROs due to its prominent characteristics such as a low power consumption, large noise margin, and ratio-less characteristic. Chen et al. proposed a full-swing clock oscillator on flexible plastic with a pseudo-CMOS and bootstrapping structure [62]. Moreover, a phase modulating capacitor (CM) is adapted to connect between the output node of stage n − 1 (P) and the bootstrap node of stage n + 1 (Q) for a square wave. Lastly, a RO is given for clock generating with a frequency of 324 kHz at VDD = 20 V which achieves an almost full swing output characteristic.

Tiwair et al. adjusted the control input signal of a pseudo-CMOS inverter, as shown in Figure 13a, where the gate of M1 (node IN1) is connected to the output of the (i − 1)-stage inverter, while the gate of M4 is connected to the output of the (i − 3)-stage (if i > 3, else (n − (i − 3))-stage) inverter of an n-stage RO [149]. As a result, the signal at the gate of M4 arrives earlier than that of M1 which compensates for the transmission delay time and improves the operation frequency.

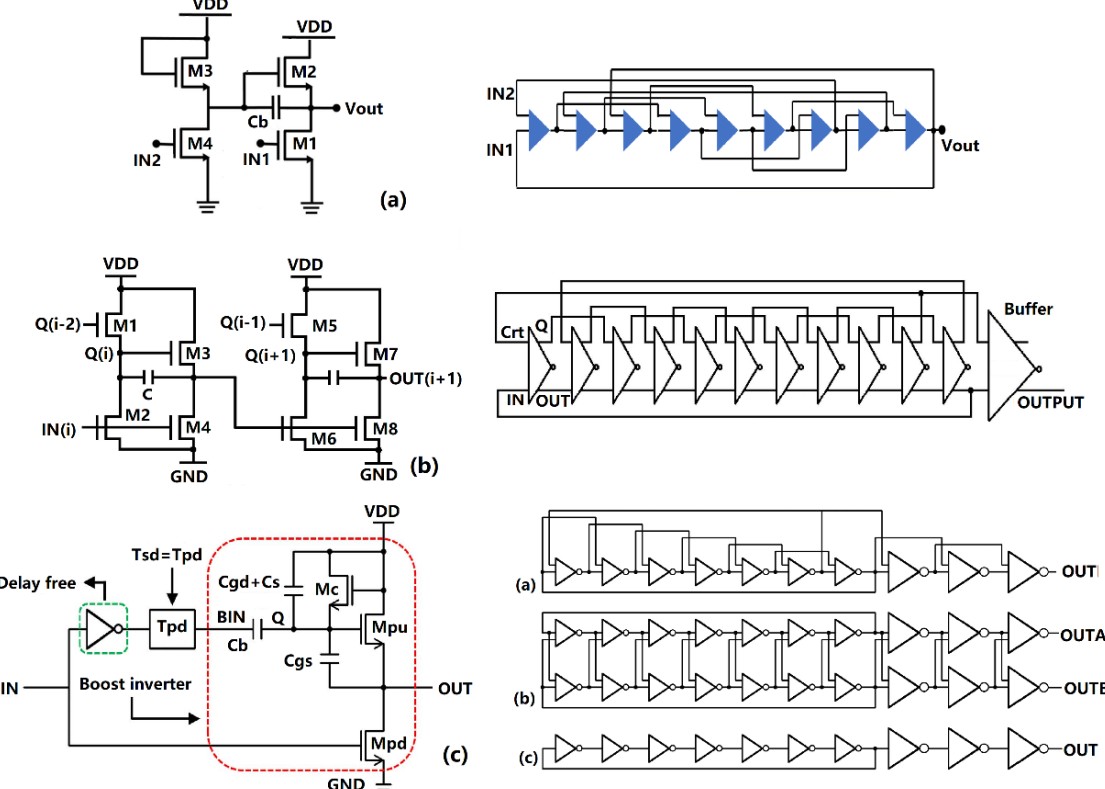

**Figure 13.** Oscillator with an optimized pseudo-CMOS inverter: (**a**) Diode-connected oscillator, (**b**) Pull-up control oscillator [79], and (**c**) Skewed delay oscillator [85].

However, a diode-connected transistor is also implemented in the pseudo-CMOS inverter. As a result, the problems related to high static power consumption cannot be completely solved. Therefore, new circuit topologies need to be developed for the RO based on oxide TFTs for achieving a high output voltage swing and low power consumption. In Figure 13b, Wu et al. proposed a RO with a pull-up control scheme in which the gate

of M1 is connected to Q(I − 2). The power consumption of the proposed RO tends to be 18% smaller than that of conventional RO, along with the increase of oscillation frequency. Furthermore, the peak-to-peak voltage of the proposed RO is promoted at least 80% more than that of a conventional RO at any oscillation frequency [63]. To improve the oscillator frequency, Xu et al. proposed a RO with a skewed delay scheme in which the pull-up TFTs is turned on or shut off before pull-down TFTs so that the skewed delay (SD) RO is at least 45% and 25% improved in frequency compared with the bootstrap RO as shown in Figure 13c [67].

Complementary structure oscillators based on p-type SnO and n-type oxides inverters are reported by Li et al. [104], consisting of a flexible oxide-TFT-based CMOS ring oscillator fabricated on a polyimide foil substrate with an oscillation frequency of 18.4 kHz. The relationship curves between peak-to-peak voltage, frequency, and bending curvature have been realized. The detailed performance of the representative inverter and oscillator of oxide-based TFTs is described in Table 2.

**Table 2.** Summarized performances for the representative inverter and oscillator of oxide-based TFTs.

| Pixel Circuits | NO. of Devices | Device Type | Structure | Circuit Type | Stages | Frequency |
|---|---|---|---|---|---|---|
| Reference [60] | 2T | CNT + IGZO | Complementary | Inverter + oscillator | 51 | 1.96 kHz |
| Reference [104] | 2T | SnO + ZnO | Complementary | Oscillator | NA | 18.4 kHz |
| Reference [61] | 2T | SnO + ZnO | Complementary | Inverter + oscillator | 5 | 8.16 kH |
| Reference [62] | 4T1C | IGZO | Diode-load pseudo-CMOS | Inverter + oscillator | 13 | 360 kHz |
| Reference [63] | 4T1C | MO | pseudo-CMOS with pull-up control | Inverter + oscillator | 11 | 132 kHz |
| Reference [64] | 2T | AlInZnSnO and InZnO | Zero-Vgs-load | Oscillator | 13 | 296 kHz |
| Reference [65] | 2T | IGZO | Diode-load | Inverter | NA | NA |
| Reference [66] | 8T2C | IGZO | Active inductor (aL) | Oscillator | NA | 5–31 Hz |
| Reference [150] | 2T | CNT+IGZO | Complementary | Inverter + oscillator | 7 | 12.3k Hz |
| Reference [151] | 2T | LTPO | Complementary | Inverter | NA | NA |
| Reference [152] | 2T | LTPO | Complementary | Inverter | NA | NA |

Amplifier

The operational amplifier is the most important module which is used to amplify signals in analog circuits. However, there are various design challenges for oxide-based TFTs realizing operational amplifiers. First, the transconductance of metal oxide TFTs is not high enough because the electrical mobility of metal oxides is much lower than that of crystalline silicon [153]. Second, it is difficult to obtain high output impedance in amplifier design due to the lack of a p-type TFTs. Some operational amplifiers based on metal oxide TFTs have been reported [154,155]. In [154], positive feedback and pseudo-CMOS technology were presented to boost the output impedance of an operational amplifier. Two-stage Cherry-Hooper topology has also been used to increase the voltage gain and gain-bandwidth product (GBWP) of amplifiers based on metal oxide TFTs [156]. In addition, compensating for the threshold voltage shift of n-type TFTs can increase the phase margin to improve stability [155]. In fact, the gain of an amplifier can also be increased if the equivalent transconductance of the amplifier is improved.

Chen et al. proposed a transconductance-enhancement topology amplifier based on oxide TFTs in Figure 14a [157]. This kind of amplifier includes two stages, one is a differential input circuit and feedback module and the other is a differential-to-single-ended converter, of which the voltage gain is 29.54 dB and the unity-gain frequency is 180.2 kHz with 21.5° PM (phase margin). The measured 3 dB bandwidth (BW) is 9.33 kHz and the gain-bandwidth product (GBWP) is calculated to be 279.9 kHz. Nevertheless, the power consumption is relatively high and noise experiment is not included.

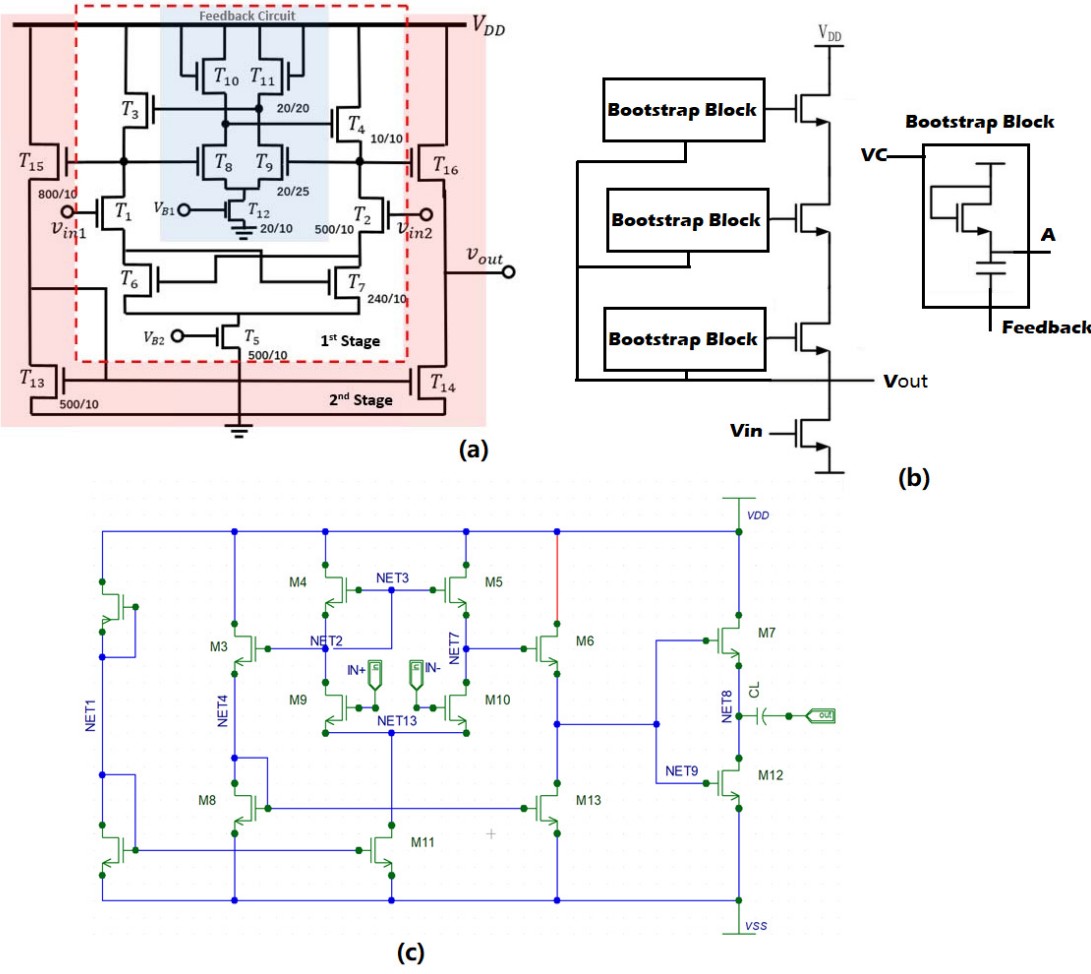

**Figure 14.** Amplifier of oxide-based TFT: (**a**) Transconductance-enhancement amplifier [157] (Reprinted with permission from ref. [157]. Copyright 2022 IEEE) and (**b**) Low noise amplifier. (**c**) LTPO amplifier.

A low noise amplifier in Figure 14b reported by Fan et al. with capacitor bootstrap structure is presented [158]. Due to the bootstrap structure, the input-referred noise and the transconductance (gm1) of the driving transistor (T1) can be suppressed. The experiment result shows that the amplifier achieves about 38.3 μV rms in a band of 1–200 Hz for the overall equivalent input noise and 32 dB gain.

LTPO can also be adapted to realize high performance CMOS OA. Rahaman et al. demonstrated the low power and high gain OA with LTPO technology leading to low power consumption, high speed, and a high degree of integration as shown in Figure 14c [159]. The whole circuit can be subdivided into two stages. The first stage comprises the input terminal, current mirror, and differential to single-ended converter stage, and the second stage is the output buffer stage. This achieves a high gain of 50.7 dB, fc of 200 kHz, fug of 7 MHz, a very high GBWP of 68.5 MHz, and small circuit area (0.06 mm$^2$).

Unlike CMOS OAs, oxide-based TFT OAs are for special occasions, like sensing systems, human health detection systems, biomedical fields, and large area optical systems [160]. Hence the performance of TFT OAs, such as gain, BW, PM, and speed, is not the primary pursuit; how to employ suitable OAs in different fields is more important. Kim et al. used a suitable amplifier for the sensing system [150]. Because of the sensor signal with low frequency, the amplifier is with a 24 dB gain, 2.5 kHz cut-off frequency (fc), and 25 kHz unity gain frequency (f0). Since TFT OAs are also applied in mobile and flexible fields, power consumption is also concerned. The detailed performances of the representative amplifier of oxide-based TFTs are described in Table 3.

**Table 3.** Summarized performances for the representative amplifier of oxide-based TFTs.

| Pixel Circuit | Topology | VDD [V] | Av [dB] | BW [kHz] | PM [Deg.] | Power [mW] | Stages |
| --- | --- | --- | --- | --- | --- | --- | --- |
| Reference [150] | CMOS | 20 | 24 | 22.5 | NA | NA | 1 |
| Reference [154] | Pseudo-CMOS | 5 | 22.5 | 5.6 | −15 | 0.16 | 2 |
| Reference [155] | Positive feedback | 6 | 19 | 25 | −70 | 6.78 | 1 |
| Reference [157] | Positive feedback | 15 | 29.54 | 9.33 | −21.5 | 5.07 | 1 |
| Reference [158] | Bootstrap structure | 20 | 26.8 | 40 | −37 | 1.53 | 1 |
| Reference [159] | CMOS | ±20 | 50.7 | 200 kHz | −92 | 0.6 | 2 |
| Reference [160] | Positive feedback | 6 | 36 | 3.2 | −32 | 0.42 | 2 |

Other Analog Circuits

In order to explore more practical applications, some researchers have been trying to employ oxide-based TFTs for ADC, rectifier, voltage-to-frequency converter, and Schmitt triggers. Garripoli et al. fabricated an a-IGZO asynchronous delta-sigma modulator on foil for accurately transforming analog sensor signals as shown in Figure 15a [82]. Compared to organic TFTs integrated with ΣΔ ADC, this modulator exhibits high performance, such as more than 19× larger BW, 9× lower power consumption, and higher SNR in a 60× larger BW. Papadopoulos et al. proposed a flexible RFID-ready ADC comprising a total of 1394 IGZO TFTs and 31 capacitors. This ADC is for printed NTC sensors to enable on-skin and sensor tag applications, as shown in Figure 15b [83].

Flexible oxide-based TFT rectifiers have been demonstrated for NFC operation by Tiwari et al. [161]. Four kinds of rectifiers fabricated on a flexible PEN substrate were compared with frequency response and amplitude sweep output, as shown in Figure 15c. A field-effect rectifier (FER) is fabricated with short-circuiting of the gate and drain structure (diode-connected TFT) as shown in Figure 15d [162]. The two-terminal rectifier with input (D + G electrode in TFT) and output (S electrode in TFT) electrodes shows a rectification ratio of $3.5 \times 10^6$ in DC measurement and maximum peak amplitude up to 20 V in AC measurement. It can steadily operate at a frequency up to 10 MHz. Below 1 MHz, the rectifier output curve can be used to distinguish the input signal transition. Voltage-to-frequency converter (VFC) and Schmitt trigger of oxide-based TFTs are analyzed by Xu [160]. VFC includes an integrator, TFT, and Schmitt trigger. The Schmitt trigger uses a pseudo PMOS structure as shown in Figure 15e, consisting of two n-type transistors, two pseudo-PMOS transistors, and one inverter. This VFC can achieve a 1.5% linearity error, 1 kHz/V tuning sensitivity, and 109 μW power consumption.

### 3.2.2. Digital Logic Circuits

A logic gate is a basic block in a digital integrated circuit. Simple logic gates consist of transistors, like the AND-OR-NOT gate. High and low voltage levels can represent the logical "true" and "false" or a binary 1 s and 0 s to realize logic operation. Conventional Si CMOS digital ICs are the most mature and popular technology while the TFTs digital ICs are suitable for the flexible and large-area field. Unipolar logic gates have weak robustness and high power consumption due to the uniformity and continuous on-state of devices. Optimization and improvement have been persistently implemented for higher performance unipolar logic gates as shown in Figure 16 [163,164]. Since oxide-based TFTs have low $V_{th}$ and off-current, oxide logic gates have the potential to downscale the power supply and consumption as mobility increases. On the other hand, the structure of TFTs are another important factor. CP TFTs with minimum or negligible parasitic overlap between gate and source/drain could provide faster logic circuits.

With optimized logic gates, digital circuits of oxide-based TFTs have been demonstrated from SSI circuits to VLSI circuits, such as D and JK flip-flops, counter, a level shifter, RFID, and digital flexible IC. D flip-flop with memory function and two stable states is the most basic logic unit which constitutes a variety of sequential logic circuits. Therefore, D flip-flop has been widely used in digital systems and computer applications, such as the digital signal register, shift register, frequency, waveform generator,

and so on. Figure 17a [164] shows a unipolar D flip-flop with internal feedback control. Figure 17b,c [77] present complementary D and JK flip-flops with p-SnO and n-IGZO TFTs.

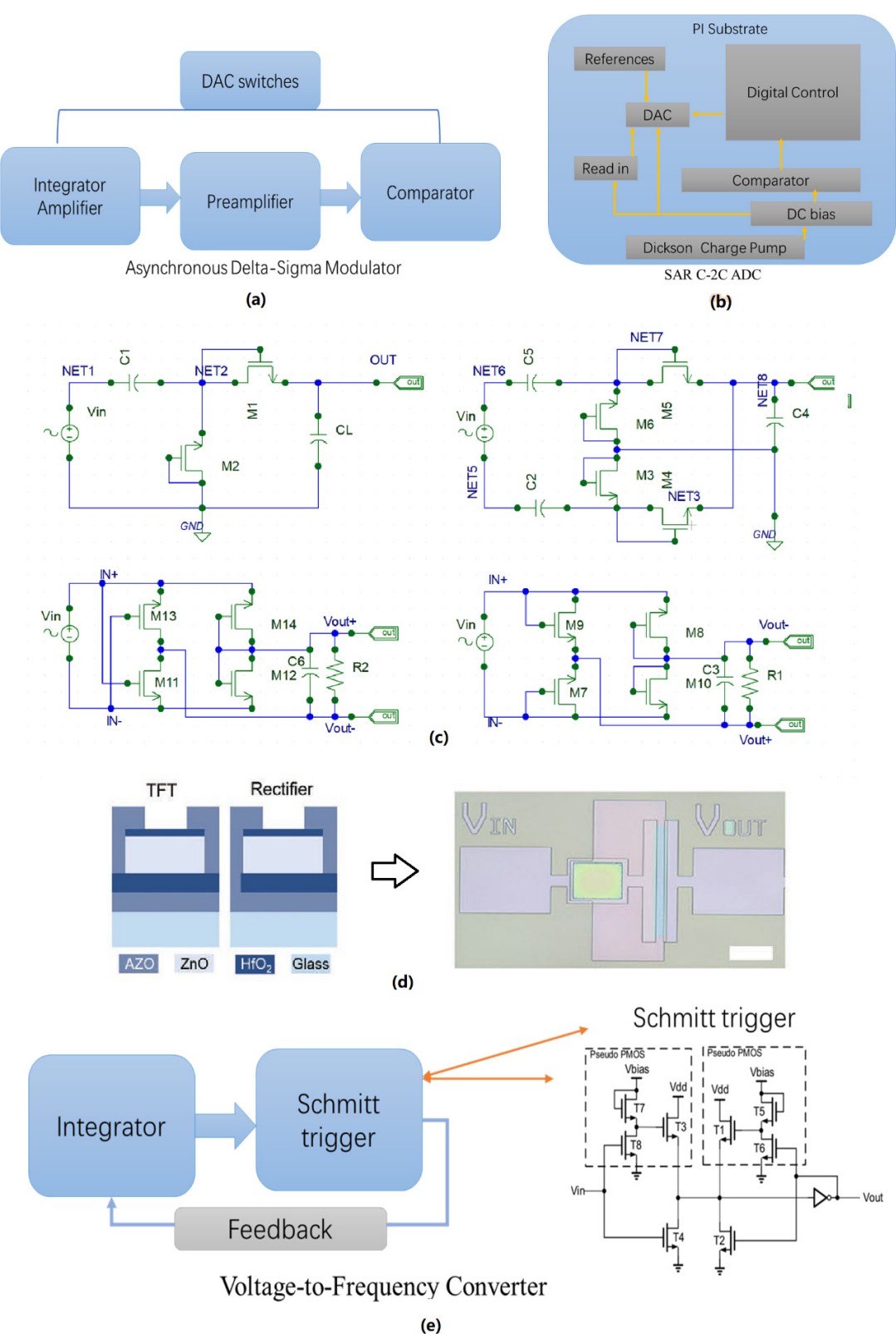

**Figure 15.** Other analog circuits based on oxide TFTs: (**a**) Diagram of the asynchronous delta-sigma modulator, (**b**) Diagram of SAR C-2C ADC, (**c**) Four kinds of rectifiers, (**d**) Field-effect rectifiers [163], and (**e**) Diagram of voltage-to-frequency converter and Schmitt trigger.

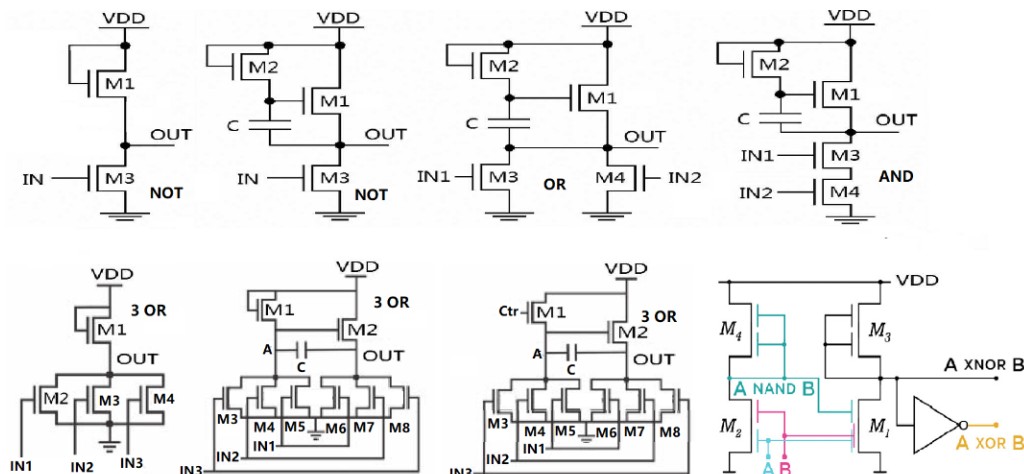

**Figure 16.** Basic logic circuit with oxide TFTs [163,164] (Reprinted with permission from ref. [164]. Copyright 2022 IOP Publishing).

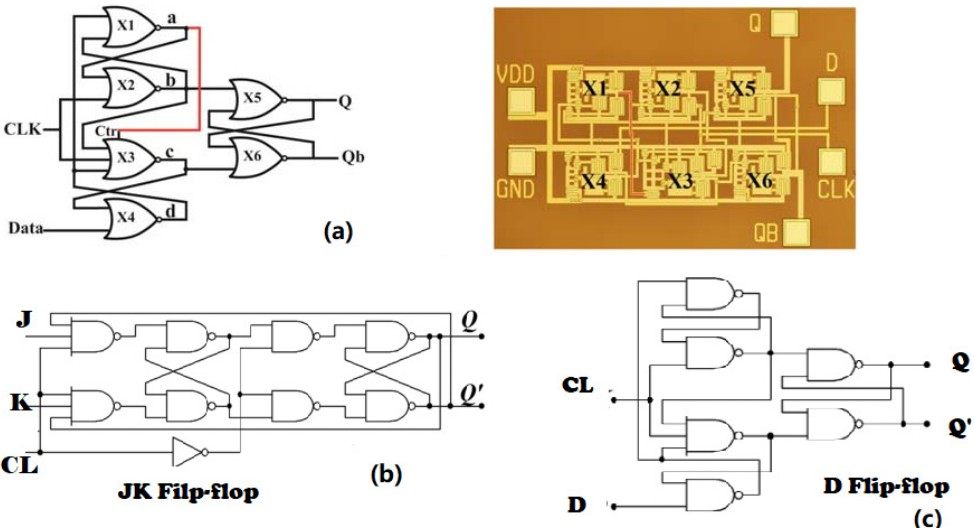

**Figure 17.** (**a**) Internal feedback control D flip-flop [164], (**b**) Complementary JK flip-flop, and (**c**) Complementary D flip-flop.

RFID technology has been widely applied in logistics, transportation management, financial security, and door access control systems. Wu et al. proposed RFID logic integrated circuits with Manchester-encoded data transmission as shown in Figure 18a [163]. It consists of basic logic circuit modules, such as a ring oscillator, a 5-bit counter, two 2–4 decoders, a 16 bit ROM, and a Manchester encoder. All the modules in the circuit are only built by the bootstrapped NOT gate and NOR gate. Flexible digital ICs with more complex structures are demonstrated with oxide-based TFTs. Ozer et al. presented a flexible binary neural network (BNN) IC with 3028 TFTs and 1461 resistors for odor classification as shown in Figure 18b [78]. This sensor processor with three-layer architecture has a 22 kHz operating frequency and only 1.1 mW power. Another flexible digital IC is a hardwired machine learning processing engine for the computation of integrated smart systems with 1000 logic gates (2084 TFTs and 1048 resistors) proposed by [79], as shown in Figure 18c. A machine learning algorithm is imported into the engine for sweat odor classification. Although the performance of oxide-based TFTs IC is not as good as CMOS IC, it is believed that more and more FTs ICs have been employed in domain-specific applications like flexible systems, fast-moving consumer goods (FMCGs), healthcare, wearables, and the Internet of Things (IoT).

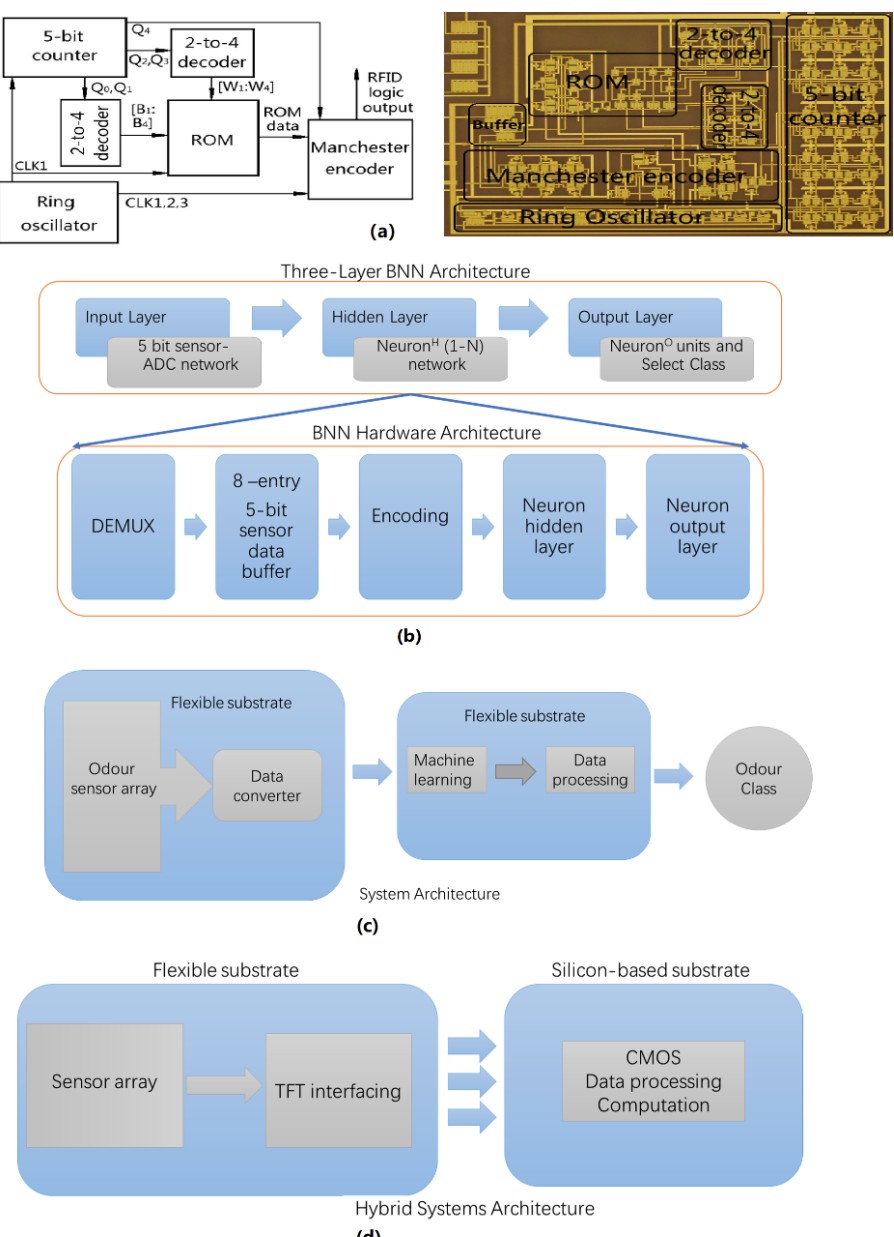

**Figure 18.** (**a**) Schematic diagram and optical image of RFID circuit [163] (Reprinted with permission from ref. [163]. Copyright 2022 INSTITUTION OF ENGINEERING AND TECHNOLOGY), (**b**) Diagram of BNN and hardware architectures, (**c**) Diagram of the system architecture of the natively flexible processing engine, and (**d**) Diagram of the system architecture and conceptual implementation of large-area sensor acquisition hybrid systems.

Hybrid sensor systems combining large-area TFT arrays with silicon-CMOS ICs for sensing and computation have been reported by [80] as shown in Figure 18d. Because sensor arrays need a large number of interfaces to connect CMOS ICs, CMOS IC is unable to satisfy this demand due to its limited pins. Hence, large-area TFT arrays are essential for sensors and CMOS ICs interface. CMOS ICs could obtain the value of each sensor by modulating the frequency or address of large-area TFT arrays so that the corresponding signal will be read out. Hybrid sensor systems give full play to TFT and CMOS technology, which is suitable for robustness and domain-specific application. Supposing there is a breakthrough in TFT performance, hybrid systems maybe the way of transition in the process of TFTs replacing CMOS, or they would be the mainstream technique of TFT digital circuit systems.

## 4. Summary and Outlook

This invited review discussed the latest strategies of oxide-based TFTs for applications, such as displays, analog circuits, and digital circuits. Researchers are persistently engaging not only in optimizing device structure and performance but also in upgrading and improving the circuit function, cost, robustness, area, and power for applications. For displays, pixel circuits of LCDs have been optimized with memory structure for a self-refreshing method to reduce the response time and power consumption. In OLEDs, internal and external compensation are the key techniques for high resolution and large area displays. The strategies of internal compensation are only for $V_{th}$ of the driving TFTs, while external compensation can include $V_{th}$, mobility, OLED degradation, and luminescence non-uniformity through the sensor TFT and optical device. The analog PWM driving method may be the main driving method for Micro-LEDs. There are few pieces of literature on analog driving methods with oxide-based TFTs. Through analysis of the PWM driving method of LTPS TFTs, oxide-based TFTs also satisfy the demand of the circuit. The strategies presented here may be further extended to other emerging display technologies, such as colloidal quantum dot LEDs [165–167], perovskite LEDs [168–171], and colloidal quantum well LEDs [172–176].

In addition to displays, this review also discusses the latest application on analog and digital circuits of oxide-based TFTs, such as the invertor, oscillator, amplifier, digital basic logic circuit, and digital processor. Because of the unipolar TFT, optimized structures of a circuit designed by researchers are discussed. The processor of oxide-based TFTs with the most complicated structure is demonstrated for sensor signal computation, making greatly potential for applications on digital ICs. Moreover, to further improve the performance of the circuit, LTPO is widely used in the field described above, realizing the complementary CMOS circuit of TFTs. The structure and performance could be optimized and improved. Despite the cost and complicated process, LTPO technology is a promising technology as the requirements for the TFTs circuits keep getting higher.

Over the past ten years, the performance of oxide-based TFTs have been enhanced step-by-step. To date, there are still many challenges hindering the development of commercial productions including robustness, reliability, and uniformity in large areas. Particularly, robustness means as the degradation occurs, we can still promise the full performance of the device by employing different device structures for different applications. For example, when in flexible substrates, the vertical structure is more appropriate due to its stress and defect suppression. While in limited substrate size, the DG structure could obtain more driving current. Reliability means to what extent devices can be sustained in adverse environmental conditions, such as strong bias stress, humidity, as well as high and low temperatures. In addition to developing higher performance materials and structures to improve reliability, new structures of driving circuits are also necessary for adapting to environmental conditions, such as differential structure, pseudo-CMOS, and complementary structure circuits. Finally, uniformity means the size of electronic we could fabricate without any mismatch on TFTs with no distortion and degradation on operating circuits. Uniformity could be achieved by improving the fabricated processes and upgrading equipment. However, the degradation of $V_{th}$ and mobility of TFTs is inevitable; new compensation circuits are required to solve these degradations such as internal and external compensation technology and optical compensation technology. After solving the above-mentioned issues, the prospect for mass production of oxide-based TFTs will be much brighter.

**Author Contributions:** L.Z., W.X., W.W. and B.L. conceived the idea; L.Z., H.Y., C.L., J.C. and B.L. wrote the paper, L.Z., M.G., H.G., W.W. and B.L. advised the paper. All authors have read and agreed to the published version of the manuscript.

**Funding:** This research was funded by National Natural Science Foundation of China (Grant No. 61874046), National Natural Science Foundation of China (Grant No. 62104265), Science and Tech-

nology Program of Guangdong Province (Grant No. 2021A0505110009), Innovation and Technology Fund (Grant No. GHP/006/20GD).

**Data Availability Statement:** Data available in a publicly accessible repository.

**Conflicts of Interest:** The authors declare no conflict of interest.

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
