# Peer review of "Strategies for Applications of Oxide-Based Thin Film Transistors"

_electronics, doi:10.3390/electronics11060960_

Round 1

Reviewer 1 Report

The authors have presented a novel and interesting work on oxide based thin film transistors (TFTs). The manuscript comprehensively states the device configurations for TFTs and up-to-date applications of these devices in thin film integrated circuits. While the properties and applications of TFTs are mentioned in an extensive method in the manuscrpit, there are points to be improved related to both the content and style of the text which are indicated below:

(1) Figure 1 should be improved. It is hard to distinguish the words on the layers. The size and font of the letters should be unified. Also, (c) in Figure 1 is written in capital. It should be corrected. 

(2) The resolution of all figures should be improved and unified. The words on the figures are mostly indistinguishable especially for Figure 9 and 14. 

(3) It would be beneficial to mention the properties of electron mobility and band gap in oxide based TFTs in comparison to other TFTs in the introduction section.

(4)It would be useful to mention which physical deposition methods are used to fabricate the oxides to be used as TFTs.

Reviewer 2 Report

This work summarizes the application of oxide-based thin film transistor which is comprehensive and helpful for the following research. I suggest it be accepted after minor revision.

(1) There are some minor grammar errors, please double check the language.

(2) Please simply describe the problem associated with the application of oxide-based thin film transistor and the corresponding strategies. 

Reviewer 3 Report

The review article titled “Strategies for Applications of Oxide-Based Thin Film Transistors” by Zhang et al. deals with a review of oxide based TFTs, their design, implementation and some strategies for improving their display as part of the IC technology. The article also reviews some of the future developments of the oxide-based TFT technology. Even though the amount of content is large, I have following comments that raise valid concerns to recommend the article for publication:

  1. The entire article has very large amount of spelling, grammatical and syntax errors that makes it a random collection of ideas and concepts and thoughts and lacks scientific soundness.
  2. The introduction section alone uses close to 100 references, which is way too much to streamline the flow of the content for the reader, especially within 70 lines! Even though the authors wanted to keep most of these references, they should have thoroughly written the Introduction section to set a stage for the sections on design, implementation and applications of the TFTs. The basic flow starting from general to specific is totally missing in the Introduction section.
  3. Figure 1f and top half of 1a are not even discussed in section 2. As in case of Introduction, the basic flow of information is missing. The authors should have started by discussing what TFTs are followed by how they are implemented and then discussing what the major configurations are, how they are different and why one is preferred over the other.
  4. Figure 2 has two different monitors having two different display pictures, which cannot be compared. For displays to be compared, the authors should have used same pictures on both the monitors and then highlighted the differences. In general, section 3 has the same issues as section 2, lot of information with no flow from general to specific, topics hopping from one technology or technique to another with a lot of syntax, grammatical and spelling errors. Many figures not even cited properly.
  5. Overall, the manuscript lacks the focus and scientific depth that is required of a review article.

Round 2

Reviewer 3 Report

The authors have made significant modifications to the manuscript as per the suggestions.